# Asymmetry between the two acidic patches dictates the direction of nucleosome sliding by the ISWI chromatin remodeler

Robert F Levendosky, Gregory D Bowman*

TC Jenkins Department of Biophysics, Johns Hopkins University, Baltimore, United States

**Abstract** The acidic patch is a functionally important epitope on each face of the nucleosome that affects chromatin remodeling. Although related by 2-fold symmetry of the nucleosome, each acidic patch is uniquely positioned relative to a bound remodeler. An open question is whether remodelers are distinctly responsive to each acidic patch. Previously we reported a method for homogeneously producing asymmetric nucleosomes with distinct H2A/H2B dimers (Levendosky et al., 2016). Here, we use this methodology to show that the Chd1 remodeler from Saccharomyces cerevisiae and ISWI remodelers from human and Drosophila have distinct spatial requirements for the acidic patch. Unlike Chd1, which is equally affected by entry- and exit-side mutations, ISWI remodelers strongly depend on the entry-side acidic patch. Remarkably, asymmetry in the two acidic patches stimulates ISWI to slide mononucleosomes off DNA ends, overriding the remodeler's preference to shift the histone core toward longer flanking DNA.
DOI: https://doi.org/10.7554/eLife.45472.001

*For correspondence:
gdbowman@jhu.edu

**Competing interests:** The authors declare that no competing interests exist.

## Introduction

In eukaryotic cells, ATP-dependent chromatin remodelers play key roles in defining the chromatin landscape by assembling, disassembling, and repositioning nucleosomes on DNA (*Becker and Workman, 2013*). The first chromatin remodeling family discovered, SWI/SNF, was identified for its ability to promote gene activation by disrupting nucleosomes (*Hirschhorn et al., 1992*; *Kwon et al., 1994*; *Peterson and Herskowitz, 1992*). In contrast, Chd1 and ISWI remodelers play an opposing role, reducing cryptic transcription in gene bodies by stimulating nucleosome assembly and organizing nucleosomes into evenly-spaced arrays (*Ito et al., 1997*; *Lusser et al., 2005*; *Smolle et al., 2012*). These two classes of remodelers appear to differ primarily through their responses to available DNA flanking the nucleosome. SWI/SNF enzymes can slide nucleosomes on top of transcription factors bound to flanking DNA, thereby dissociating them, whereas Chd1 and ISWI remodelers preferentially shift nucleosomes away from transcription factors, preserving their occupancy (*Dechassa et al., 2010*; *Kang et al., 2002*; *Li et al., 2015*; *Nagaich et al., 2004*; *Nodelman et al., 2016*).

Directional sliding by Chd1 and ISWI is achieved by coupling DNA sensing to DNA translocation by a conserved ATPase motor (*Flaus et al., 2006*; *Nodelman et al., 2017*; *Yang et al., 2006*). The ATPase motor can shift nucleosomes by acting at either of the two symmetrically related superhelix location 2 (SHL2) sites (*McKnight et al., 2011*; *Schwanbeck et al., 2004*; *Zofall et al., 2006*). At each SHL2 site, the ATPase motor always shifts DNA toward the nucleosome dyad; therefore, action at each site results in DNA movement in opposite directions. Relative to the SHL2 site where the motor acts, flanking DNA that is pulled onto the histone core, referred to as entry-side DNA, is

located on the opposite edge of the nucleosome disk, whereas DNA exiting the core is on the same edge at the ATPase motor but a full superhelical turn away. As directional sliding factors, Chd1 and ISWI remodelers show highest sliding activity when DNA is longer and exposed on the entry side and shorter or inaccessible on the exit side (*Nodelman et al., 2016*; *Yang et al., 2006*). Chd1 and ISWI remodelers possess a similarly folded DNA-binding domain (DBD) that interacts with flanking DNA (*Dang and Bartholomew, 2007*; *Grüne et al., 2003*; *Nodelman et al., 2017*; *Ryan et al., 2011*; *Yamada et al., 2011*). For the DBD to reach entry-side DNA with the ATPase motor simultaneously bound at SHL2, the remodeler therefore has to span across the width of the nucleosome disk.

In addition to DNA flanking the nucleosome, remodelers are also sensitive to other epitopes, presented by the histone core. Recently, the ATPase subunit of the human ISWI remodeler, called SNF2h, and the related ACF complex were shown in two studies to be sensitive to disruption of the nucleosome 'acidic patch' (*Dann et al., 2017*; *Gamarra et al., 2018*), a distinctive cluster of eight acidic residues on H2A and H2B that participate in chromatin condensation (*Dorigo et al., 2003*; *Luger et al., 1997*; *Shogren-Knaak et al., 2006*). While the acidic patch is also used by many chromatin factors for nucleosome recognition and binding (*Kalashnikova et al., 2013*; *McGinty and Tan, 2015*), Chd1 was found to only have a modest (~2 fold) defect in sliding nucleosomes with acidic patch mutations (*Levendosky et al., 2016*). One key difference in these studies was that for the experiments with Chd1, only the acidic patch on the entry H2A/H2B dimer was mutated, whereas for the ISWI studies, the acidic patch was mutated on both entry and exit dimers. Therefore, a possible explanation for this discrepancy was that Chd1 might rely more heavily on the exit side acidic patch, which was not tested. A different but related question is whether the ISWI remodeler SNF2h depends equally on both acidic patches, or is more sensitive to mutations on entry or exit H2A/H2B. Although the two acidic patches are related to each other by 2-fold symmetry, each is in a distinct location relative to the SHL2 site where the remodeler ATPase motor acts. Thus, determining whether entry or exit acidic patches are more deleterious is an important question because it should reveal architectural aspects of how remodelers engage key nucleosome epitopes.

We previously showed that, using the Widom 601 nucleosome positioning sequence, hexasomes can be produced in an oriented fashion, where the sole H2A/H2B dimer preferentially deposits on the so-called TA-rich side (*Levendosky et al., 2016*). Since nucleosomes can be generated from addition of H2A/H2B dimers to hexasomes, oriented hexasomes allows for the homogeneous production of nucleosomes having different mutations or modifications on each H2A/H2B dimer (*Levendosky et al., 2016*). Here, we use this methodology for producing nucleosomes having one or both acidic patches mutated in a position-specific fashion, to probe whether the Chd1 remodeler from yeast and ISWI remodelers from human and *Drosophila* show greater sensitivity to mutations on the entry or exit side H2A/H2B dimer. Our findings indicate that ISWI and Chd1 remodelers differ in how each remodeler responds to acidic patch asymmetry in the histone core. Whereas Chd1 is similarly affected by acidic patch mutations on entry and exit H2A/H2B dimers, ISWI depends much more strongly on having a wild type acidic patch on the entry dimer. This dependence results in strongly biased sliding of asymmetric nucleosomes by ISWI, with net movement toward the side containing the wild type acidic patch. These findings suggest that in vivo, selective blocking of one acidic patch would locally redirect ISWI action to disorganize and potentially disrupt nucleosomes.

## Results

### Chd1 is similarly affected by defects in the entry and exit side acidic patch

To reveal the spatial requirements that Chd1 and ISWI remodelers may have for the acidic patch, we generated several nucleosomes with acidic patch mutations (APMs, *Figure 1A*) on one or both sides. To generate homogeneous pools for each combination of mutant/wild type H2A, oriented hexasomes were first made with an H2A/H2B dimer that possessed either a wild type acidic patch (gray) or the APM (red) (*Figure 1B*). Then, addition of either wild type or APM H2A/H2B dimers to each of these hexasomes produced the four possible combinations of wild type/APM nucleosomes. Since

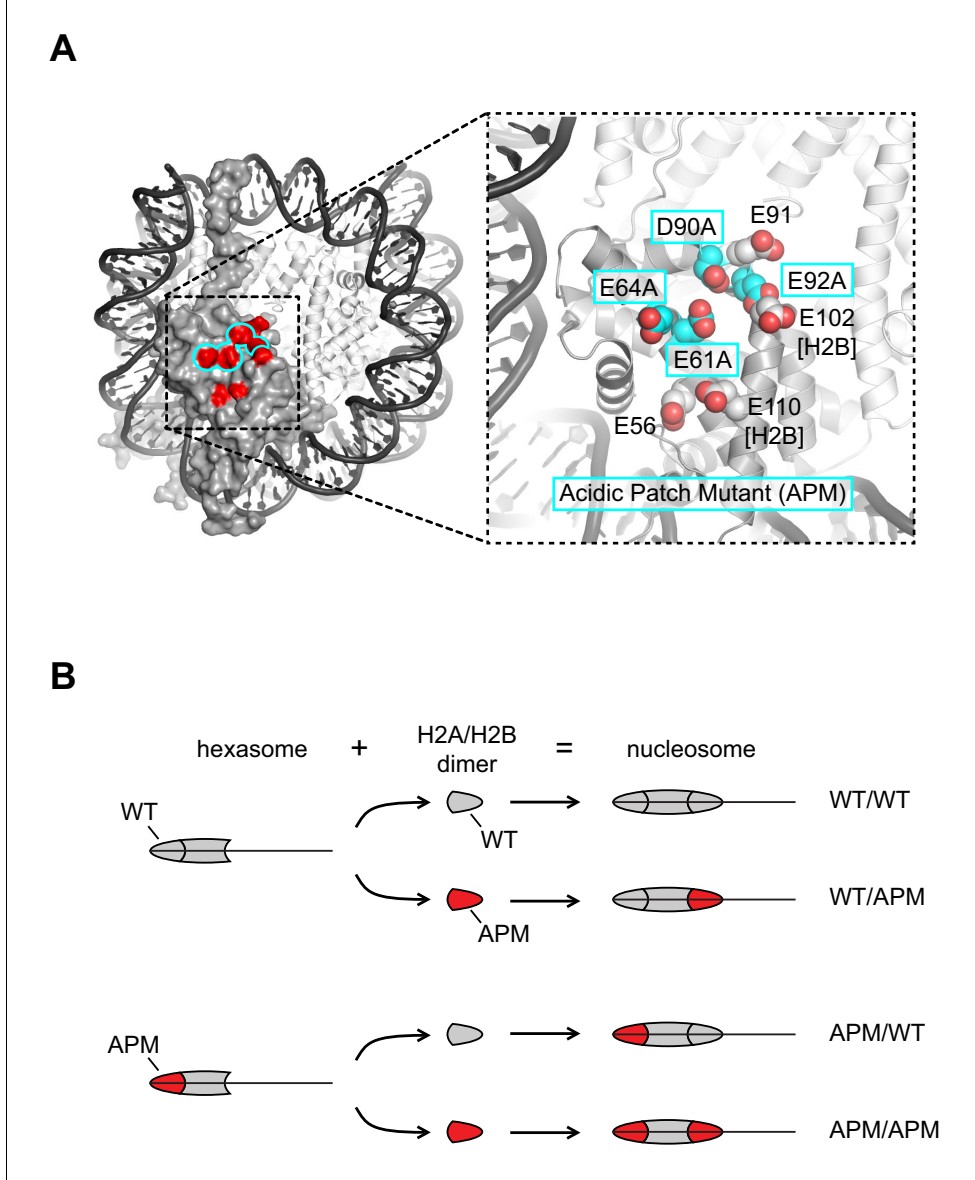

**Figure 1.** Design of nucleosomes with acidic patch mutations (APMs). (**A**) The acidic patch of the nucleosome. In the overview, the surface is shown for one H2A/H2B dimer, with eight residues comprising the acidic patch highlighted in red. The acidic patch mutation (APM) consists of four alanine substitutions on H2A (E61A/E64A/D90A/E92A), highlighted in cyan. The structure shown is PDB code 1KX5 (*Davey et al., 2002*). (**B**) Strategy for making asymmetric nucleosomes using oriented hexasomes. As previously shown (*Levendosky et al., 2016*), hexasomes made with the Widom 601 (*Lowary and Widom, 1998*) preferentially lack an H2A/H2B dimer on the TA-poor side (here, the right side). Addition of either a wild type (gray) or APM H2A/H2B dimer (red) converts hexasomes to nucleosomes, allowing for production of all four combinations of H2A/H2B dimers with and without the APM. The locations of the H2A/H2B dimers and the H3/H4 tetramer on the Widom 601 is depicted in *Figure 1—figure supplement 1*.

DOI: https://doi.org/10.7554/eLife.45472.002

The following figure supplement is available for figure 1:

**Figure supplement 1.** Arrangement of histone contacts with the Widom 601.

DOI: https://doi.org/10.7554/eLife.45472.003

Chd1 and ISWI preferentially shift mononucleosomes to more central positions on short DNA fragments (*Stockdale et al., 2006*; *Yang et al., 2006*), we designed the nucleosomes to be end-positioned, with 80 bp of flanking DNA on one side and zero bp on the other. We previously found that the asymmetry of the 601 sequence affected the sliding behavior of Chd1, with a preference for shifting the histone octamer toward the TA-poor side of the 601 sequence (*Winger and Bowman, 2017*). Therefore, we made two distinct sets of end-positioned nucleosomes, with the long 80 bp flanking DNA either extending from the TA-poor side (by convention here, to the right, referred to as a 0N80 construct), or extending from the TA-rich side (to the left, referred to as an 80N0 construct). For each nucleosome, the two H2A/H2B dimers were either both wild type (WT/WT), both mutated (APM/APM), or contained the acidic patch mutation specifically on the left (APM/WT) or the right (WT/APM).

To investigate how an APM on different sides affects Chd1 activity, we performed native gel sliding assays to monitor the rate of nucleosome centering (*Figure 2*). For all eight nucleosomes, Chd1 shifted histone octamers away from the DNA ends, consistent with nucleosome centering activity. For both 0N80 and 80N0 constructs, the nucleosomes containing the double APM/APM were shifted most slowly, whereas single APM nucleosomes were only slightly slower than wild type. To better visualize how different nucleosome substrates affected rates, the disappearance of end-positioned nucleosome was plotted over time and fit to single exponential progress curves (*Figure 2B*). Compared to reactions of WT/WT and APM/APM nucleosomes, progress curves for reactions with single APM H2A/H2B dimers showed intermediate rates, with faster rates when the APM dimer was on the same side as the 80 bp flanking DNA compared to when APM was on the zero bp side.

In agreement with our previous findings (*Levendosky et al., 2016*), APM on the entry-side dimer (adjacent to the 80 bp DNA) had a rather modest effect, slowing down the sliding reaction by ~2 fold (*Figure 2C,D*). Here, we find that sliding was also slowed down (~3 fold) by an APM on the exit side H2A/H2B (adjacent to the zero bp DNA side). Interestingly, APM on both sides appeared to have an additive effect, resulting in 7- to 10-fold slower sliding. As we anticipated, the sliding rates for the 0N80 nucleosomes (TA-poor adjacent to 80 bp flanking DNA, *Figure 2C*, blue bars) was faster than 80N0 nucleosomes (TA-rich adjacent to 80 bp flanking DNA, gray bars). Yet despite these rate differences, the relative slow-down due to APM on one or both sides was remarkably similar, and therefore relatively unbiased by the underlying DNA sequence (*Figure 2D*). The additive effect of APM implies that either a single remodeler can sense both acidic patches, or possibly that two remodelers on opposite sides of the nucleosome can communicate in a way that is sensitive to the acidic patch. Previous experiments showed that disrupting both acidic patches (APM/APM) more dramatically reduced sliding activity of SNF2h (~200 fold) compared with ACF (~10 fold) (*Gamarra et al., 2018*). Therefore, the impact of the double APM nucleosomes on Chd1 activity was similar to the reduction in the rate of nucleosome centering observed for ACF.

## ISWI cannot center asymmetric nucleosomes containing a single APM

To investigate whether ISWI was similarly dependent on both sides of the nucleosome, and how impactful single APM-containing H2A/H2B dimers would be, we performed similar native gel sliding experiments using human SNF2h and *Drosophila* ACF (*Figure 3*). In agreement with previous results (*Dann et al., 2017*; *Gamarra et al., 2018*), both SNF2h and ACF shifted APM/APM nucleosomes significantly more slowly than WT/WT nucleosomes. As observed with Chd1, SNF2h was sensitive to the 601 sequence orientation, sliding both WT/WT and APM/APM nucleosomes more readily in the 0N80 orientation, with the TA-poor side of 601 on the side adjacent to the 80 bp flanking DNA (*Figure 3A*).

Interestingly, with asymmetrically placed APMs, both SNF2h and ACF showed different behaviors depending on whether the APM was on the 80 bp or 0 bp side. When the APM was on the 0 bp side, nucleosomes initially moved toward the center (up the gel, migrating more slowly) but then appeared to shift back toward the DNA ends (down the gel) (*Figure 3A,B*, 0N80 APM/WT and 80N0 WT/APM). After moving up the gel, the return of nucleosomes to the lower, end position coincided with the emergence of free DNA (highlighted with asterisks). For SNF2h, similar increases in free DNA were also observed for single APM-containing nucleosomes on the 80 bp side. These nucleosomes (80N0 APM/WT and 0N80 WT/APM) were different in that they did not yield any slower migrating species, suggesting that SNF2h was unable to shift these toward more central

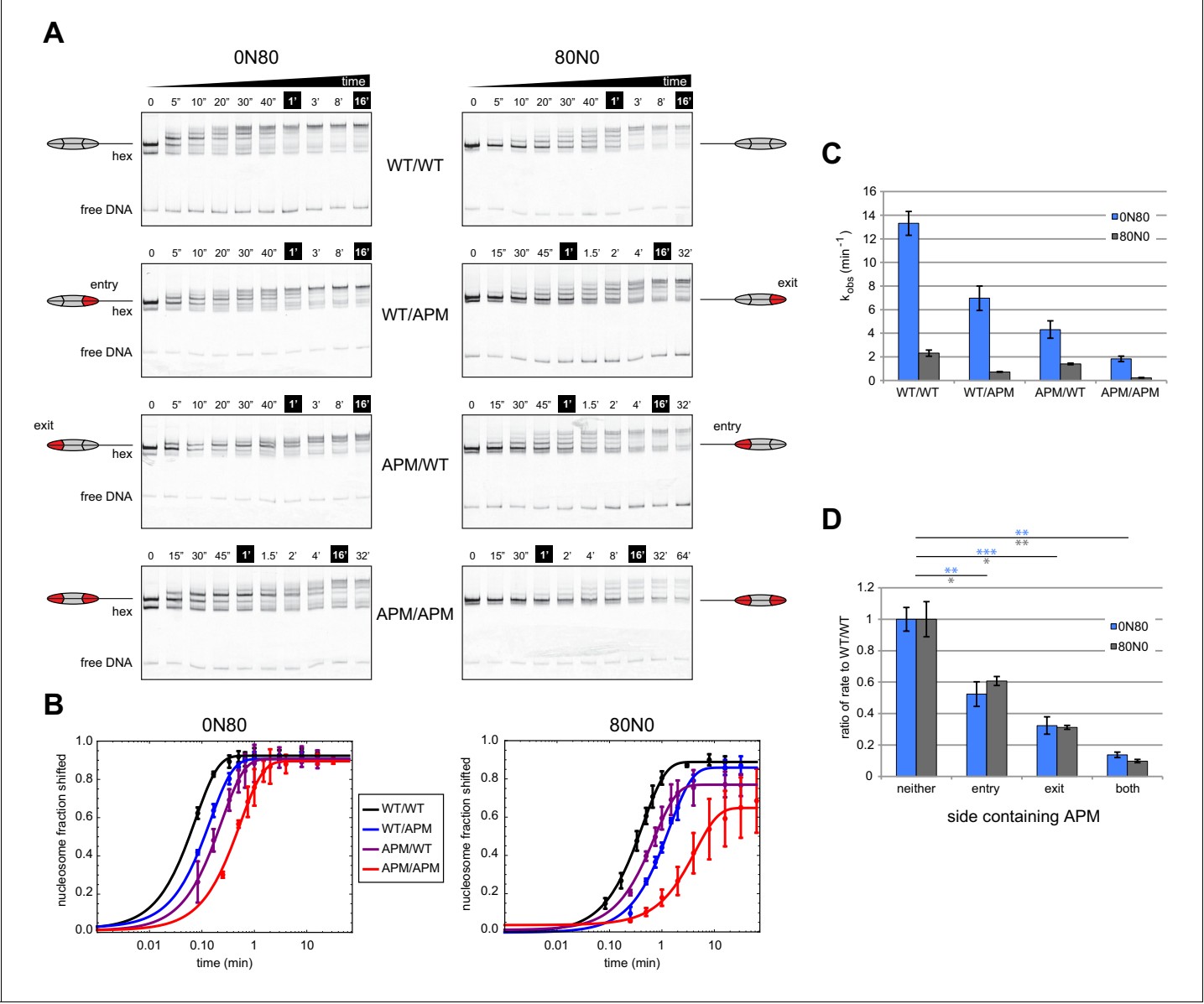

**Figure 2.** Mutations in the acidic patch slow down but do not disrupt centering ability of Chd1. (**A**) Native gel nucleosome sliding assays using Chd1, with all four arrangements of APM and WT H2A/H2B dimers. Depending on which side the flanking DNA was on, the unique APM dimers were either on entry or exit sides, as indicated. The positions of the nucleosome cartoons indicate end-positioned nucleosomes in the gel, with the lower band representing residual hexasomes. Nucleosome centering is evidenced by slower migration in the gel. Reactions contained 40 nM hexasome, 60 nM H2A/H2B dimer, 200 nM Chd1, and 100 µM ATP. Note the different time series used (indicated above each gel), which helped capture sliding intermediates given the different reaction rates. Results are representative of three independent replicates. (**B**) Quantified data from (**A**) plotting the disappearance of end-positioned nucleosomes, overlaid with single exponential fits. The end-positioned band intensity was normalized to the total band intensity within each lane. Error bars show standard deviations from three replicates. (**C**) Mean observed rates ± standard deviations from fits obtained in (**B**). (**D**) Relative impact of APMs on Chd1 remodeling rates. Remodeling rates for WT/WT 0N80 and 80N0 were each scaled to 1. Rates for nucleosomes containing APM substitutions were scaled relative to WT/WT made with the same DNA construct. P-values *$\leq$0.02; **<0.005; ***<0.0005.
DOI: https://doi.org/10.7554/eLife.45472.004

locations. Interestingly, for 80N0 nucleosomes with an APM on the 80 bp side (APM/WT), SNF2h instead produced a faster migrating species expected for hexasomes (*Figure 3A*).

When quantified relative to nucleosome bands, the free DNA species clearly increased over time for both ACF and SNF2h on APM/WT 0N80 nucleosomes (*Figure 3C*). The same strong increase was not observed for WT/WT nucleosomes, indicating that the asymmetry of the acidic patch was

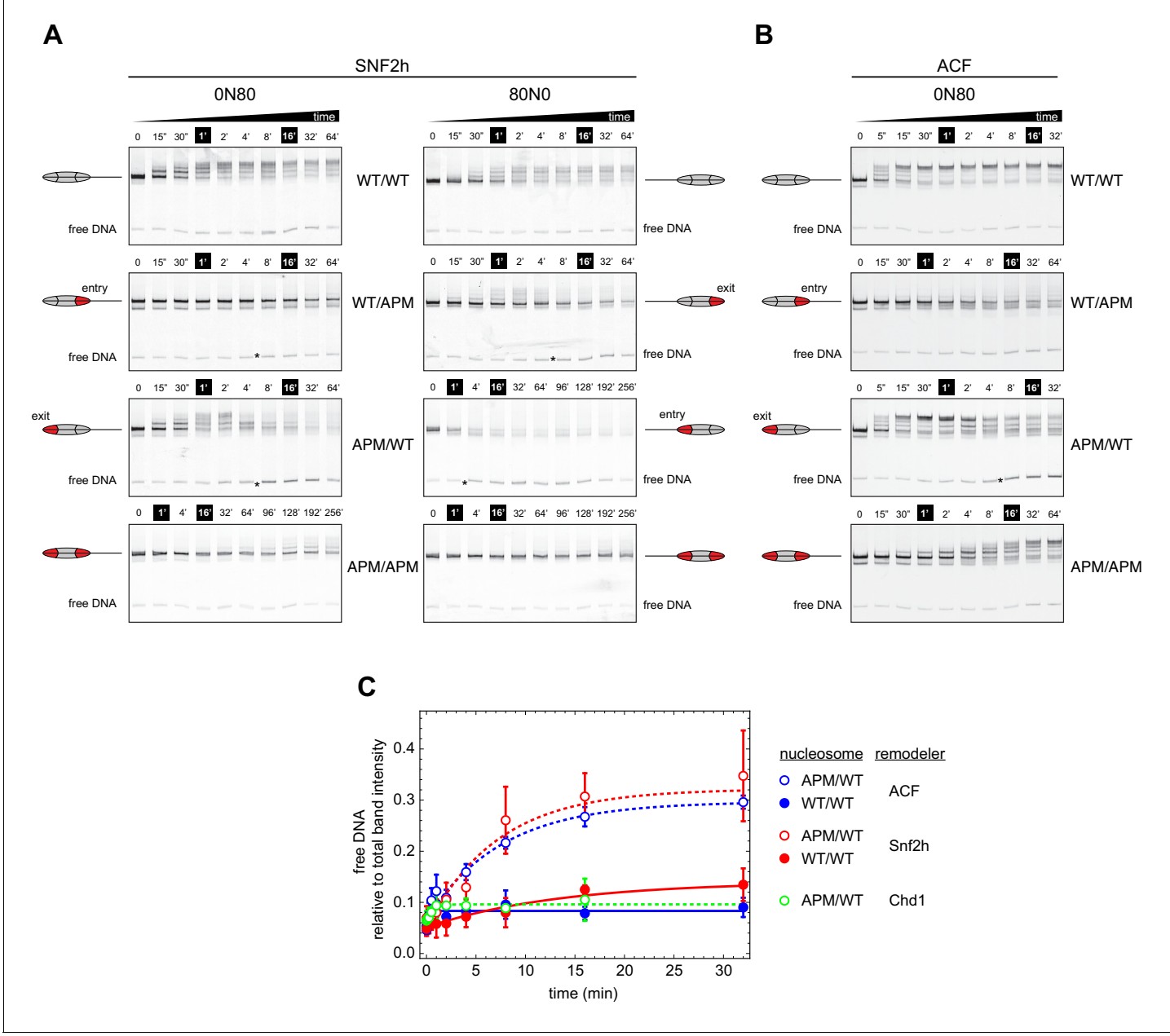

**Figure 3.** Asymmetry between the two nucleosome acidic patches disrupts nucleosome centering by ISWI remodelers. (**A**) Native gel nucleosome sliding assays using SNF2h, with all four arrangements of WT and APM H2A/H2B dimers for 0N80 and 80N0 nucleosomes. Nucleosome centering corresponds to slower migration of nucleosome species. Asterisks highlight noticeable increases in free DNA. Remodeling reactions contained 40 nM hexasome, 80 nM H2A/H2B dimer, 1 μM SNF2h, and 2 mM ATP. Results are representative of two independent replicates. (**B**) Native gel nucleosome sliding experiments of 0N80 nucleosomes as in (**A**), but using *Drosophila* ACF. Remodeling reactions contained 40 nM hexasome, 70 nM H2A/H2B dimer, 100 nM ACF, and 2 mM ATP. Results are representative of two or more replicates. (**C**) ISWI remodelers but not Chd1 stimulate release of DNA from 0N80 nucleosomes with APM/WT histone cores. Shown is the accumulation of free DNA normalized to total band intensity from native gel experiments. All reactions tracking free DNA contained 40 nM hexasome, 70 nM H2A/H2B dimer, and 2 mM ATP with either 1 μM Snf2h, 100 nM ACF, or 200 nM Chd1. The mean and standard deviation of three or four replicates is plotted with single exponentials fits.
DOI: https://doi.org/10.7554/eLife.45472.005

The following figure supplements are available for figure 3:

**Figure supplement 1.** For the ACF complex, association of the Acf1 subunit tracks with nucleosome sliding activity.
DOI: https://doi.org/10.7554/eLife.45472.006

**Figure supplement 2.** At sub-saturating concentrations, ACF shifts nucleosomes with asymmetric acidic patch mutations toward DNA ends.
DOI: https://doi.org/10.7554/eLife.45472.007

essential. A similar increase in free DNA was not observed for Chd1, reinforcing the conclusion that Chd1 and ISWI respond differently to asymmetric APM/WT nucleosomes.

Given the common behaviors of ACF and SNF2h in sliding the asymmetric APM nucleosomes, one potential concern we had was whether the ATPase subunit of ACF was dissociated from Acf1. If dissociated, it would not be surprising for the two ATPases without their auxiliary subunit to exhibit similar behavior. Produced commercially, the *Drosophila* ACF complex was affinity purified by virtue of a FLAG tag on the Acf1 subunit. We reasoned that if the ACF complex was intact, FLAG-Acf1 would be stably associated with the ATPase subunit, allowing both subunits to be pulled down by anti-FLAG antibodies. To test this idea, we incubated ACF with anti-FLAG beads, with the expectation that retention of the intact ACF complex on anti-FLAG beads would remove all nucleosome sliding activity from the flow-through. As a positive control, ACF was passed through a spin filter without anti-FLAG beads, and the flow-through showed nucleosome sliding activity as expected (*Figure 3—figure supplement 1A*). In contrast, when an ACF sample was incubated with anti-FLAG beads, the flow-through showed no detectable activity (*Figure 3—figure supplement 1B*). As another control, the anti-FLAG beads were pre-treated with FLAG peptide before incubation with ACF. This pre-treatment resulted in a similar level of nucleosome sliding activity as the no-beads control, indicating that the FLAG-binding ability of the beads was necessary for retention of activity (*Figure 3—figure supplement 1C*). These control experiments indicate that Acf1 was tightly associated with the ATPase subunit, supporting the notion that the striking sliding behavior of APM/WT 0N80 nucleosomes were characteristics of the full ACF complex.

One observation we made in the course of these experiments was that the centering-then-decentering behavior of ACF on APM/WT 0N80 nucleosomes appeared to be concentration dependent. With excess ACF over nucleosomes (100 nM ACF, 40 nM nucleosomes), ACF initially shifted end-positioned nucleosomes toward a more central position, followed by a redistribution toward a DNA end (*Figure 3B*, APM/WT). Yet with a limiting amount of ACF (10 nM ACF, 40 nM nucleosomes), only an initial nucleosome shift toward the center was apparent with a similar time course (*Figure 3—figure supplement 1*). However, this apparent concentration dependence instead may simply reflect a much slower rate of equilibration with undersaturating ACF, because with extended time points (up to 90 hr), the migration pattern clearly showed a redistribution of nucleosomes back toward end positioning (*Figure 3—figure supplement 2*).

These results show that both human SNF2h and *Drosophila* ACF are strikingly different from yeast Chd1 in how they remodel these asymmetric nucleosomes. Moreover, the appearance of free DNA with APM/WT 0N80 nucleosomes suggests that SNF2h and ACF can destabilize nucleosomes containing a single APM.

## SNF2h slides mononucleosomes with asymmetrically mutated acidic patches off DNA ends

We suspected that for nucleosomes containing a single APM, the apparent destabilization and inability to maintain histone octamers in a central location may reflect biased, directional movement of nucleosomes. The native gel sliding experiments were consistent with the histone octamer preferentially moving toward the side possessing the wild type acidic patch. To investigate this possibility, we tracked nucleosome sliding at high resolution using histone mapping. This method labels a single cysteine histone mutant with the photo-activatable cross-linker 4-azidophencyl bromide (APB), which forms site-specific cross-links to DNA that can stimulate cleavage of the DNA backbone. Here we labeled H2B(S53C) with APB, which results in nicking one strand of the DNA duplex ~54 bp from the nucleosome dyad and can be used to report on the position of the histone octamer (*Kassabov and Bartholomew, 2004*). Since both DNA strands are labeled (FAM and Cy5), DNA cross-linking from both copies of H2B(S53C) allows nucleosome positions to be determined even when the histone octamer is shifted 'off the end' of the DNA fragment, where one H2B(S53C) site loses contact with DNA (*Patel et al., 2013*; *Zofall et al., 2006*).

Due to limited quantities of available ACF, all histone mapping was carried out with SNF2h, using all eight combinations of 0N80 and 80N0 nucleosomes (*Figure 4* and *Figure 4—figure supplement 1*). For the symmetric nucleosomes, which contained either two wild type acidic patches or two APM sites, the histone octamers were shifted by SNF2h to more central locations on the DNA fragment (*Figure 4A,D* and *Figure 4—figure supplement 1A,D*). For the asymmetric nucleosomes, however, the pattern of nucleosome repositioning was quite different. For both 0N80 and 80N0 nucleosomes,

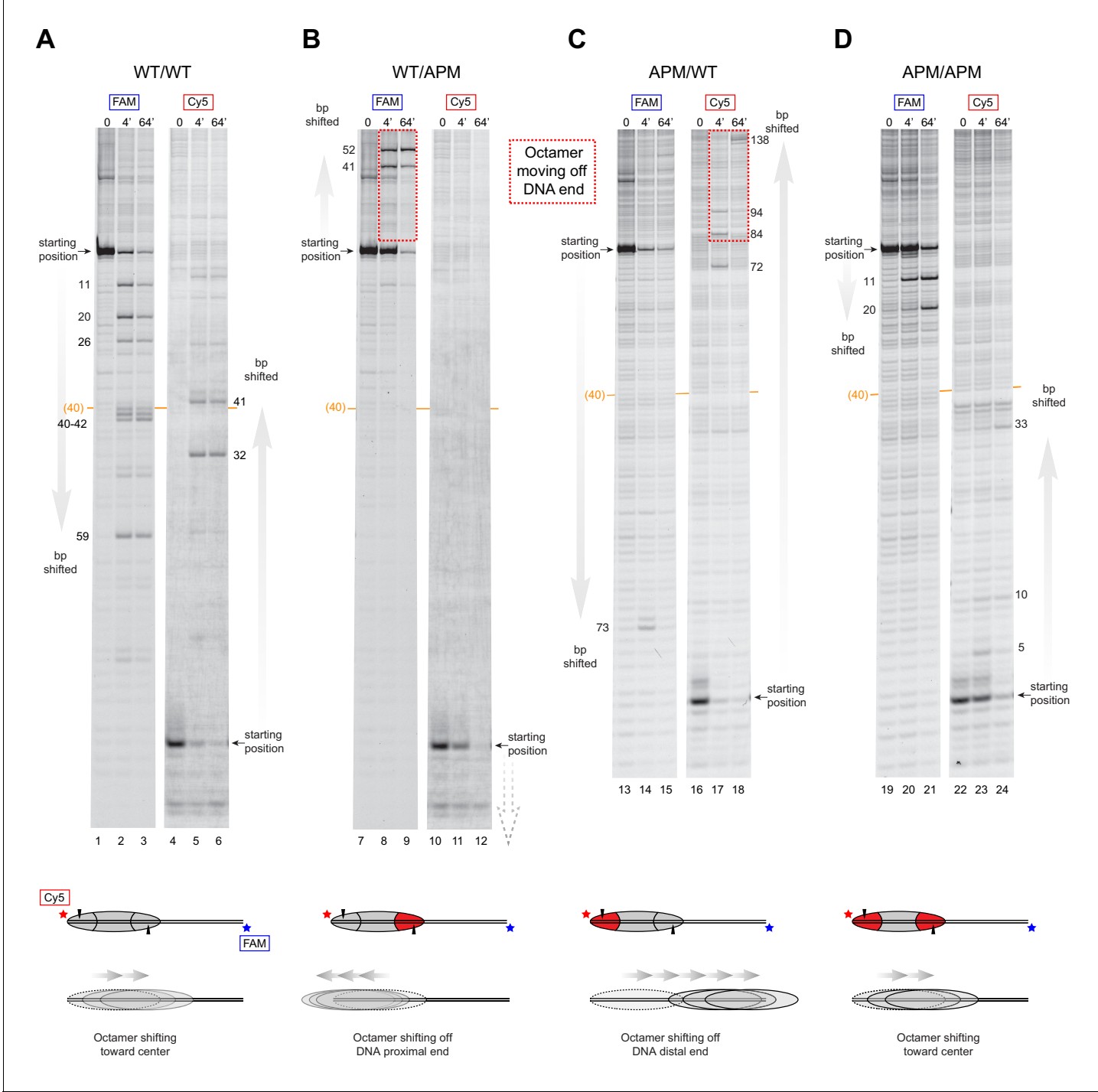

**Figure 4.** SNF2h slides histone cores with asymmetric acidic patch mutations off DNA ends. Histone mapping of SNF2h sliding reactions, performed with 0N80 nucleosomes having the four combinations of wild type and APM H2A/H2B dimers: (**A**) WT/WT, (**B**) WT/APM, (**C**) APM/WT, and (**D**) APM/ APM. Shown are scans of urea denaturing gels, which report on the locations of the histone octamer based on sites of H2B(S53C) cross-linking. Two scans are shown for each nucleosome, based on Cy5 and FAM labels on the top and bottom DNA strands, respectively. Numbering beside the gels indicates the distances (bp) the histone octamer shifted relative to the starting position, with the midpoint of the 80 bp flanking DNA highlighted with a central orange bar. Each nucleosome was generated by addition of 200 nM wild type (gray) or APM (red) H2A/H2B dimers to 100 nM wild type or APM hexasome. Cartoon schematics below each gel indicate the composition and orientation of APM and wild type H2A/H2B dimers, the sites of H2B(S53C) cross-linking (black triangles), and interpretations of sliding reactions. Sites where the histone octamer moves off DNA ends are highlighted by dotted red boxes. Sliding reactions contained 1 μM SNF2h and 2 mM ATP, and reactions were quenched at 0, 4, and 64 min. These results are

*Figure 4 continued on next page*

*Figure 4 continued*

representative of two independent experiments. Analogous experiments using 80N0 nucleosomes are shown in *Figure 4—figure supplement 1*. Extended gels are shown in *Figure 4—figure supplement 2* and *Figure 4—figure supplement 3*.

DOI: https://doi.org/10.7554/eLife.45472.008

The following figure supplements are available for figure 4:

**Figure supplement 1.** SNF2h slides histone cores with single acidic patch mutations off DNA ends regardless of Widom 601 orientation.

DOI: https://doi.org/10.7554/eLife.45472.009

**Figure supplement 2.** Extended gel images of histone mapping experiments using WT/WT and WT/APM nucleosomes.

DOI: https://doi.org/10.7554/eLife.45472.010

**Figure supplement 3.** Extended gel images of histone mapping experiments using APM/WT and APM/APM nucleosomes.

DOI: https://doi.org/10.7554/eLife.45472.011

the histone octamer always shifted in the direction of the H2A/H2B dimer containing the wild type acidic patch, regardless of flanking DNA length. On 0N80 nucleosomes, SNF2h shifted WT/APM octamers off the zero side by up to 52 bp (*Figure 4B*). For the APM/WT nucleosomes, where the wild type and mutated acidic positions are swapped, SNF2h shifted the histone octamer in the opposite direction, across the entire length of the flanking 80 bp (*Figure 4C*). The same behavior of preferentially shifting the histone octamer toward the side with the wild type acidic patch was observed for 80N0 nucleosomes (*Figure 4—figure supplement 1*). For all asymmetric APM/WT and WT/APM nucleosomes, the histone octamer shifted farther than the available flanking DNA, which means that portions of the DNA-binding surface of the histones became exposed as the DNA end was pulled to a more internal position on the nucleosome. In each case, the DNA end was shifted up to ~50 bp past the canonical edge of the nucleosome, which corresponds to the internal SHL2 site where the ATPase motor acts. These results are consistent with the native gel sliding experiments (*Figure 3*), and suggest that movement of the histone octamer off DNA ends was destabilizing and likely responsible for hexasome formation and free DNA.

The movement of the octamer 'off the end' of the DNA, means that SNF2h continued to pull DNA onto the nucleosome even after no entry-side flanking DNA was available. SWI/SNF remodelers are well known to be insensitive to DNA flanking the nucleosome, and characteristically slide histone octamers off DNA ends (*Saha et al., 2005*; *Zofall et al., 2006*). In contrast, however, ISWI remodelers typically shift histone octamers to more central locations on DNA fragments, requiring entry-side DNA for movement (*Stockdale et al., 2006*; *Yang et al., 2006*). Our observation that SNF2h slides these octamers off DNA ends suggests that for ISWI, the importance of the entry-side acidic patch outweighs the necessity of available entry-side flanking DNA.

## Discussion

Nucleosome spacing activity, characteristic for Chd1 and ISWI remodelers, arises from an ability to sense and respond to DNA flanking the nucleosome. To generate evenly spaced nucleosomes, these remodelers act on both sides of the nucleosome, with a preference for shifting the histone core toward the side with longer available DNA (*Stockdale et al., 2006*; *Yang et al., 2006*). In this study we found that DNA length sensing by Chd1 did not appear affected when one of the two acidic patches of the nucleosome was disrupted. In contrast, asymmetric presentation of the two acidic patches appeared to dictate the direction of nucleosome sliding by ISWI, overriding the enzyme's normal requirement for flanking DNA on the entry side of the nucleosome.

For both Chd1 and ISWI remodelers, activation of the ATPase motor at SHL2 correlates with engagement of the DBD on entry-side DNA (*Hwang et al., 2014*; *McKnight et al., 2011*; *Yang et al., 2006*). Notably, for the DBD to simultaneously contact flanking DNA at the entry side while the ATPase motor binds SHL2, the remodeler would have to reach across one of the two faces of the nucleosome disk, where it could encounter the acidic patch of either the entry or exit H2A/H2B dimer. Chd1 was affected to a similar extent whether a single APM was on one side or the other, and although the histone octamers were asymmetric, both APM/WT and WT/APM nucleosomes were centered by Chd1 (*Figure 2*). This behavior is consistent with stimulation of Chd1 activity when interacting with either entry- or exit-side acidic patches, implying that Chd1 may reach from SHL2 to entry DNA across either face (*Figure 5A*).

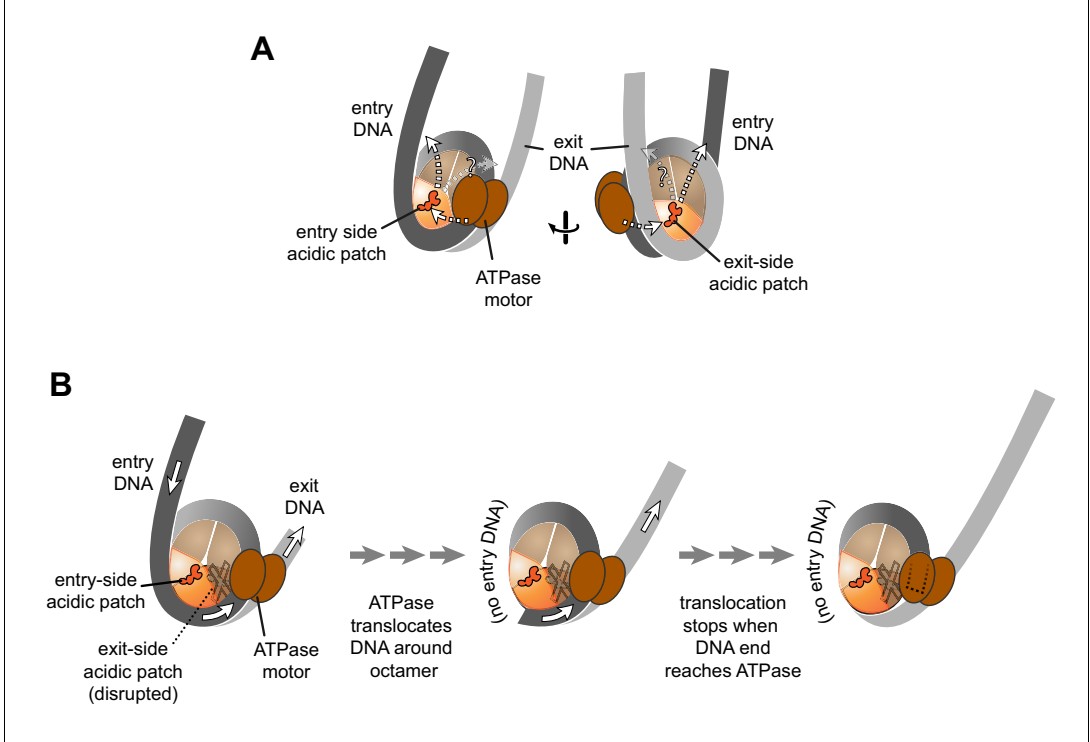

**Figure 5.** Schematics showing the spatial relationship between the remodeler ATPase motor, the two acidic patches, and flanking DNA of the nucleosome. (**A**) A cartoon representation showing the two sides of a nucleosome with a remodeler ATPase motor bound at an SHL2 site. Broken arrows show paths connecting the ATPase motor with entry or exit DNA. (**B**) An illustration of how sliding the histone core off DNA ends impacts flanking DNA relative to the ATPase motor. As for ISWI, if action of the ATPase motor on one side of the nucleosome is severely limited due to an acidic patch mutation (brown "X"), DNA will be preferentially shifted in one direction due to ATPase action on the side with the wild type acidic patch (orange). Continued action on one side, with insufficient counteraction on the other side, will eventually pull all entry side DNA completely onto the histone core. If translocation continues without the need for flanking DNA on the entry side, the end of the DNA will eventually reach the ATPase motor at SHL2 (dotted outline).

DOI: https://doi.org/10.7554/eLife.45472.012

The following figure supplement is available for figure 5:

**Figure supplement 1.** Chd1 remodelers have a longer linker connecting the ATPase motor and DNA-binding domain than ISWI.

DOI: https://doi.org/10.7554/eLife.45472.013

SNF2h and ACF, in contrast, had a strikingly asymmetric response to nucleosomes containing one APM and one wild type acidic patch. Rather than centering, the ISWI remodelers always moved asymmetric histone cores toward the side with the wild type H2A/H2B dimer, indicating that activity relied heavily on the entry side acidic patch. Interestingly, the segment between the DBD and NegC, which immediately follows the ATPase, is shorter in ISWI (~40–45 residues) compared with the corresponding segment between the DBD and C-terminal bridge of Chd1 (~75–130 residues) (*Figure 5—figure supplement 1*). The shorter connecting segment for ISWI may therefore limit the path between SHL2 and entry DNA to the nucleosome face containing the entry side H2A/H2B. This potential difference in engaging with only entry side or with both faces of the nucleosome may reflect how remodeler action at the two SHL2 sites is coordinated. For both ISWI and Chd1, two copies of the remodeler can simultaneously occupy the two SHL2 sites on opposite sides of the nucleosome (*Nodelman et al., 2017*; *Racki et al., 2009*; *Sundaramoorthy et al., 2018*). For SNF2h and ACF, however, the remodeler binds cooperatively to nucleosomes as dimers (*Racki et al., 2009*), whereas no cooperativity has been reported for Chd1. Instead, Chd1 has been shown to dynamically switch between both sides of the nucleosome, perhaps helping to ensure back-and-forth motion without an opposing remodeler (*Qiu et al., 2017*). Consistent with acting as a monomer, Chd1 is sensitive to DNA flanking both sides of the nucleosome (*Nodelman et al., 2016*), requiring entry-side DNA for tethering the remodeler to the nucleosome (*McKnight et al., 2011*) yet also

making intramolecular interactions when engaged with exit-side DNA (*Farnung et al., 2017*; *Nodelman et al., 2017*; *Sundaramoorthy et al., 2018*). Whether interactions with the acidic patch are favored when the Chd1 DBD is on entry or exit DNA is presently unclear.

For ISWI, the strong dependence on the entry side acidic patch resulted in the asymmetric histone cores shifting off DNA ends (*Figure 4* and *Figure 4—figure supplement 1*). This off-the-end movement correlated with the creation of free DNA (*Figure 3C*) and indicated that, as long as the entry-side acidic patch is available, ISWI can continue to pull DNA onto the nucleosome even after no more entry DNA is available (*Figure 5B*). Previously, the engagement of the ISWI DBD with entry DNA was considered to be a key step in remodeler activation, proposed to stimulate removal of the inhibitory NegC element from the ATPase motor (*Clapier et al., 2017*). Indeed, this idea is consistent with the recent discovery that disruption of NegC largely bypasses the need for activation by the acidic patch (*Gamarra et al., 2018*). However, our finding that SNF2h can pull DNA ends up to the internal SHL2 where the ATPase motor acts (*Figure 4*) shows significant remodeler activity in the absence of entry side DNA (*Figure 5B*). This conclusion does not conflict with the model that sufficiently long entry-side DNA stimulates remodeling activity, as previously proposed (*Hwang et al., 2014*; *Yang et al., 2006*). Instead, these findings show that the acidic patch is a more important nucleosomal epitope than flanking DNA, and that binding to entry DNA is not essential for activating ISWI remodeling activity.

The directional sliding we observed with these asymmetric nucleosomes suggests a potential means that ISWI activity might be modulated in vivo. Recent work (*Dann et al., 2017*; *Gamarra et al., 2018*) showed that, analogously to mutating the acidic patch, remodeling interference could be accomplished by addition of an acidic patch-binding peptide called LANA (*Barbera et al., 2006*). With the growing list of chromatin binding factors that interact with the acidic patch (*McGinty and Tan, 2015*), it is likely that competition for this important epitope in vivo impacts remodeler activity (*Gamarra et al., 2018*). Based on work described here, we expect that asymmetric association of an acidic patch binding factor, on only one side of the nucleosome, would temporarily disrupt the ability of ISWI remodelers to create or maintain evenly spaced nucleosome arrays. In addition, one possibility is that asymmetric blockage of the acidic patch could transform ISWI remodelers into disruptive enzymes. Remodelers like SWI/SNF that can slide mononucleosomes off DNA ends are disruptive (*Hirschhorn et al., 1992*; *Kwon et al., 1994*; *Peterson and Herskowitz, 1992*; *Ulyanova and Schnitzler, 2005*; *Ulyanova and Schnitzler, 2007*), at least in part because shifting a nucleosome into the territory of an adjacent nucleosome is destabilizing (*Dechassa et al., 2010*; *Engeholm et al., 2009*). We previously reported that a modified version of Chd1, with the DBD replaced by monomeric streptavidin, could shift biotinylated histones off DNA ends and disrupt canonical nucleosome wrapping in arrays (*Patel et al., 2013*). This showed that the ability to slide nucleosomes without regard for entry DNA is sufficient for nucleosome-disrupting activity (*Patel et al., 2013*). For remodelers like ISWI, a block on only one acidic patch would allow nucleosomes to be shifted into, rather than away from, neighboring nucleosomes, and would disrupt rather than preserve transcription factor binding to entry-side DNA. The breaking of inherent two-fold symmetry in the nucleosome can be accomplished through binding of chromatin factors, histone variants, and post translational modifications, and we anticipate that both masking and enhancement of nucleosome epitopes asymmetrically will provide unique strategies for altering the outcome of remodeling reactions.

## Materials and methods

### Key resources table

| Reagent type (species) or resource | Designation | Source or reference | Identifiers | Additional information |
| --- | --- | --- | --- | --- |
| Antibody | mouse monoclonal ANTI-FLAG M2 Affinity Gel | Millipore Sigma | Cat #:2220 | (coupled to beads) |

*Continued on next page*

*Continued*

| Reagent type (species) or resource | Designation | Source or reference | Identifiers | Additional information |
|---|---|---|---|---|
| Recombinant DNA reagent | core Widom 601 (uppercase) and flanking DNA sequences (lowercase) | *Lowary and Widom, 1998* | | 5'_gggatcctaatgaccaaggaaa gcatgattcttcacaccgagttcatcc cttatgtgatggaccctatacgcggc cgcccTGGAGAATCCCGGTGCC GAGGCCGCTCAATTGGTCGTA GacAGCTCTAGCACCGCTTAAA CGCACGTACGCGCTGTCCCCCG CGTTTTAACCGCCAAGGGGAT TACTCCCTAGTCTCCAGGCACG TGTCAGATATATACATCCTGtgcat gtattgaacagcgaccttgccggtgccag tcggatagtgttccgagctcccactctaga ggatccccgggtaccg_3' |
| Sequence-based reagent | primer: Cy5-0-601 | IDT | | 5'/5Cy5/TGGAGAATCCCGGTGCC GAGGCCGCTCAAT |
| Sequence-based reagent | primer: 601–80-FAM | IDT | | 5'/56-FAM/cggtacccggggatcctcta gagtgggagc |
| Sequence-based reagent | primer: 601–0-Cy5 | IDT | | 5'/5Cy5/CAGGATGTATATATCTGAC ACGTGCCTGGA |
| Sequence-based reagent | primer: FAM-80–601 | IDT | | 5'/56-FAM/gggatcctaatgaccaagg aaagcatgatt |
| Peptide, recombinant protein (*Saccharomyces cerevisiae*) | ScChd1$_{118-1274}$ | *McKnight et al., 2011* | | from *Saccharomyces cerevisiae* |
| Peptide, recombinant protein (*Homo sapiens*) | HsSNF2h | *Yang et al., 2006* | | from *Homo sapiens* |
| Peptide, recombinant protein (*Drosophila melanogaster*) | DmACF | Actif Motif | Cat #:31509 | from *Drosophila melanogaster* |
| Peptide, recombinant protein (*Xenopus laevis*) | XlHistone H2A | *Luger et al., 1997* | | from *Xenopus laevis* |
| Peptide, recombinant protein (*X. laevis*) | XlHistone H2A-E61A/E64A/D90A/E92A 'APM' | *Girish et al., 2016* | | from *Xenopus laevis* |
| Peptide, recombinant protein (*X. laevis*) | XlHistone H2B | *Luger et al., 1997* | | from *Xenopus laevis* |
| Peptide, recombinant protein (*X. laevis*) | XlHistone H2B-S53C | *Kassabov et al., 2002* | | from *Xenopus laevis* |
| Peptide, recombinant protein (*X. laevis*) | XlHistone H3-C110A | *Dechassa et al., 2010* | | from *Xenopus laevis* |
| Peptide, recombinant protein (*X. laevis*) | XlHistone H4 | *Luger et al., 1997* | | from *Xenopus laevis* |
| Commercial assay or kit | Thermo sequenase dye primer manual cycle sequencing kit | Affymetrix | Cat #:79260 | |
| Chemical compound, drug | NaCl | Fisher Scientific | Cat #:S641-500 | |
| Chemical compound, drug | Trizma base | Sigma | Cat #:T1503-1KG | |

*Continued on next page*

Continued

| Reagent type (species) or resource | Designation | Source or reference | Identifiers | Additional information |
|---|---|---|---|---|
| Chemical compound, drug | Boric Acid | Fisher Scientific | Cat #:A73-500 | |
| Chemical compound, drug | MgCl$_2$ | Fisher Scientific | Cat #:BP214-500 | |
| Chemical compound, drug | EDTA | Thermo Fisher | Cat #:AM9260G | |
| Chemical compound, drug | KCl | Fisher Scientific | Cat #:P217-500 | |
| Chemical compound, drug | DTT | Sigma Aldrich | Cat #:D9779 | |
| Chemical compound, drug | Sucrose | Fisher Scientific | Cat #:S5-500 | |
| Chemical compound, drug | Nonidet P-40 | Fisher Scientific | Cat #:MP1RIST1315 | |
| Chemical compound, drug | BSA | New England Biolabs | Cat #:B9000S | |
| Chemical compound, drug | Salmon Sperm DNA | Invitrogen | Cat #:15632–011 | |
| Chemical compound, drug | 40% acrylamide/bis solution 19:1 | Bio-Rad | Cat #:1610144 | |
| Chemical compound, drug | Urea | Sigma Aldrich | Cat #:U1250 | |
| Chemical compound, drug | Acrylamide | Bio-Rad | Cat #:1610101 | |
| Chemical compound, drug | Bis N,N'-Methylene-Bis-Acrylamide | Bio-Rad | Cat #:1610201 | |
| Chemical compound, drug | 2-mercaptoethanol | Sigma Aldrich | Cat #:M6250 | |
| Chemical compound, drug | 4-Azidophenacyl bromide | Sigma Aldrich | Cat #:A6057 | |
| Chemical compound, drug | Phenol:Chloroform 5:1 | Sigma Aldrich | Cat #:P1944 | |
| Chemical compound, drug | Adenosine triphosphate disodium salt hydrate | Sigma Aldrich | Cat #:A1852 | |
| Chemical compound, drug | dNTPs | Invitrogen | Cat #:10297–018 | |
| Other | Multipurpose Mini Spin Columns | BioVision | Cat # 6572–50 | |
| Other | HisPrep FF 16/10 (Nickel affinity) | GE | Cat # 28-9365-51 | |
| Other | HisTrap HP, 5 ml (Nickel affinity) | GE | Cat # 17-5248-01 | |
| Other | HiTrap SP FF, 5 ml | GE | Cat # 17-5157-01 | |
| Other | HiTrap Q FF, 5 ml | GE | Cat # 17-5156-01 | |
| Other | HiLoad 16/600 Superdex 200, prep grade | GE | Cat # 28-9893-35 | |
| Other | HiLoad 16/600 Superdex 75, prep grade | GE | Cat # 28-9893-33 | |

*Continued*

| Reagent type (species) or resource | Designation | Source or reference | Identifiers | Additional information |
|---|---|---|---|---|
| Other | HiPrep 26/10 Desalting | GE | Cat # 17-5087-01 | |
| Other | HiPrep 16/10 Q FF | GE | Cat # 17-5190-01 | |
| Other | HiPrep 16/10 SP FF | GE | Cat # 17-5192-01 | |
| Software, algorithm | ImageJ | imagej.nih.gov/ij/ | | |
| Software, algorithm | Mathematica | Wolfram | | |

## Reagent preparation

Histones, DNA and Chd1 (from *S. cerevisiae*, residues 118–1274) were prepared as previously described (*Dyer et al., 2004*; *Hauk et al., 2010*; *Levendosky et al., 2016*; *Luger et al., 1999*; *Patel et al., 2011*). Asymmetric nucleosomes were made from hexasomes and H2A/H2B dimers according to *Levendosky et al. (2016)*.

SNF2h (*Homo sapiens*) was expressed and purified as previously described (*Leonard and Narlikar, 2015*) with some modifications. SNF2h was expressed from a modified pET3 expression vector (pBH4, kindly provided by Geeta Narlikar), containing an N-terminal 6xHis tag followed by a TEV cut site. Using transformed Rosetta DE3 *E. coli* cells, cultures were grown at 37°C in 2X LB media with 1X NaCl, ampicillin (100 µg/mL) and chloramphenicol (34 µg/mL) to an optical density at 600 nm ($OD_{600}$) of ~0.3 before reducing the temperature to 18°C. Growth was continued to $OD_{600}$ of ~0.6 before inducing with 400 µM IPTG. After ~18 hr of expression, cells were harvested, flash frozen and stored at −80°C. After thawing, cells were resuspended and incubated on ice for 30 min in HisBind Buffer A (20 mM Tris-HCl pH 7.8, 500 mM NaCl, 10 mM imidazole, and 10% (v/v) glycerol) supplemented with 1 µg/mL pepstatin, 3 µg/mL leupeptin, 2 mM β-mercaptoethanol, 0.1 mM PMSF, 2.5 mM MgCl$_2$, 0.5 mM CaCl$_2$, 1 mg/mL lysozyme, and 10 mg/mL DNase I. Cells were sonicated and the lysate clarified by centrifugation at 45,000 x g at 4°C for 30 min. Clarified lysate was passed over three tandem 5 mL HisTrap HP columns (GE 17–5248) pre-equilibrated in HisBind buffer A. After extensive washing, protein was eluted with a gradient of HisBind B (20 mM Tris-HCl pH 7.8, 500 mM NaCl, 1M imidazole, and 10% (v/v) glycerol), with SNF2h eluting between 210 and 300 mM imidazole. Fractions containing SNF2h, as determined by 12% SDS-PAGE, were pooled and buffer exchanged into SEC buffer (25 mM HEPES pH 7.5, 300 mM KCl, 2 mM β-mercaptoethanol) using an Amicon Ultra 10,000 MWCO spin concentrator. Protein was further purified by passage over a HiPrep 16/10 Q FF column (GE 28-9365-43), with SNF2h in the flow through. Collected protein was concentrated to <10 mL and dialyzed (6–8 KDa MWCO) overnight at 4°C in SEC buffer, with addition of ~1 mg of TEV to cleave the His tag. After addition of imidazole to 7 mM, the cleaved pool was passed over a single 5 mL HisTrap HP column to remove TEV and uncleaved protein. SNF2h was then concentrated and further purified by passage over a HiLoad 16/600 Superdex 200 size exclusion column (GE 28-9893-35) equilibrated in SEC buffer. Based on analysis from 12% SDS-PAGE, purified SNF2h was pooled, concentrated and brought to 10% (v/v) glycerol before flash freezing in liquid N$_2$ and storing at −80°C.

*Drosophila* ACF complex was purchased from Active Motif (cat. no. 31509). To test the integrity of the ACF complex, FLAG pulldown experiments (*Figure 3—figure supplement 1*) were performed to see if the Acf1 subunit, which was N-terminally FLAG tagged, was stably associated with the Iswi subunit, which was necessary for nucleosome sliding. First, 10 µl of anti-FLAG M2 beads (millipore sigma 2220) was loaded onto a multipurpose spin column (BioVision 6572), with the suspension solution removed by centrifugation at 8000 x g for 30 s at 4°C. The beads were washed twice with 50 µl of slide buffer without BSA (20 mM HEPES-KOH pH 7.5, 100 mM KCl, 5 mM MgCl$_2$, 1 mM DTT, 5% (w/v) sucrose) followed by centrifugation. Immunoprecipitation of the FLAG tag was carried out by adding 15 µl of 100 nM ACF complex to each column and incubating while shaking for 30 min at 4°C. After immunoprecipitation, the unbound remodeler was recovered into a clean microfuge tube by centrifugation. The resulting flow through was used in nucleosome sliding reactions (described

below) containing 40 nM APM hexasome, 70 nM WT H2A/H2B dimer, 2 mM ATP and 10 nM ACF, observed by native acrylamide gel electrophoresis. To control for loss or degradation of protein on the column, an equivalent amount of ACF was added to a spin column without anti-FLAG beads (*Figure 3—figure supplement 1A*). As a separate control to ensure that loss of activity was due to specific pulldown of the FLAG epitope, anti-FLAG beads were pre-blocked with 100 µl of 1 mg/mL 3xFLAG peptide for 1 hr at 4°C. Before addition of ACF, beads were washed twice with 50 µl of slide buffer without BSA to remove unbound peptide (*Figure 3—figure supplement 1C*).

### Nucleosome sliding

Nucleosome sliding reactions monitored by native acrylamide gels were performed as described (*Levendosky et al., 2016*) with some variations. Reactions were assembled at room temperature in sliding buffer (20 mM HEPES-KOH pH 7.5, 100 mM KCl, 5 mM $MgCl_2$, 0.1 mg/mL BSA, 1 mM DTT, 5% (w/v) sucrose) beginning with the addition of 40 nM hexasome and 60–80 nM dimer, which were equilibrated for 10 min, followed by the addition of either Chd1 (200 nM), SNF2h (1 µM) or ACF (10 nM or 100 nM) and another 10 min equilibration. Reactions were initiated with addition of ATP and samples were collected at the indicated time-points by quenching 1 µL of the remodeling reaction into 8 µL of quench buffer (20 mM HEPES-KOH pH 7.5, 100 mM KCl, 0.1 mg/ml BSA, 1 mM DTT, 5% (w/v) sucrose, 25 mM EDTA, 2 µg/µl salmon sperm DNA) and immediately placed on ice. Quenched reactions (3 µL) were loaded on native gels (7% acrylamide with a 60:1 acrylamide:bis-acrylamide crosslinking ratio) and electrophoresed at 130V for 1.5 to 2 hr at 4 °C. Gels were scanned on a Typhoon 5 multiphase scanner (GE) and band intensity was quantified using ImageJ software. For Chd1 reactions, the disappearance of starting material was fit to single exponential decay using Mathematica software (Wolfram).

Histone mapping was performed as previously described (*Levendosky et al., 2016*) with slight modifications. Reactions (50 µL) were assembled in sliding buffer (20 mM Tris-HCl pH 7.5, 50 mM KCl, 5 mM $MgCl_2$, 0.1 mg/mL BSA, 1 mM DTT, 5% sucrose (w/v)) with 100 nM hexasome, 200 nM H2A/H2B dimer, and 1 µM SNF2h and equilibrated for 10 min at room temperature. Reactions were initiated with addition of 2 mM ATP and quenched at the indicated time-points by adding 100 ul of quench buffer (20 mM Tris-HCl pH 7.5, 50 mM KCl, 5 mM $MgCl_2$, 0.1 mg/mL BSA, 1 mM DTT, 5% sucrose (w/v), 25 mM EDTA and 1 µg/µL salmon sperm DNA) and transferring to ice. The remainder of the procedure was conducted as previously published (*Levendosky et al., 2016*).

## Acknowledgements

We appreciate discussions and helpful suggestions from Ilana Nodelman and other members of the Bowman lab. This work was supported by the NIH (R01-GM084192 and R01-GM113240).

## Additional information

### Funding

| Funder | Grant reference number | Author |
| --- | --- | --- |
| National Institutes of Health | R01-GM084192 | Robert F Levendosky Gregory D Bowman |
| National Institutes of Health | R01-GM113240 | Robert F Levendosky Gregory D Bowman |

The funders had no role in study design, data collection and interpretation, or the decision to submit the work for publication.

### Author contributions

Robert F Levendosky, Conceptualization, Formal analysis, Investigation, Visualization, Methodology, Writing—original draft, Writing—review and editing; Gregory D Bowman, Conceptualization, Formal analysis, Supervision, Funding acquisition, Visualization, Writing—original draft, Writing—review and editing

#### Author ORCIDs
Robert F Levendosky  http://orcid.org/0000-0002-5101-0810
Gregory D Bowman  https://orcid.org/0000-0001-8025-4315

#### Decision letter and Author response
Decision letter https://doi.org/10.7554/eLife.45472.017
Author response https://doi.org/10.7554/eLife.45472.018

## Additional files

### Supplementary files
• Transparent reporting form
DOI: https://doi.org/10.7554/eLife.45472.014

### Data availability
All data generated in this study are included in the manuscript.

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
