## [Decision Letter]

Thank you for submitting your article "For the SNF2h chromatin remodeler, a requirement for the entry-side H2A acidic patch dominates over flanking DNA" for consideration by *eLife*. Your article has been reviewed by three peer reviewers, including Jerry L Workman as the Reviewing Editor and Reviewer #1, and the evaluation has been overseen by Jessica Tyler as the Senior Editor. The following individual involved in review of your submission has agreed to reveal their identity: Tom Owen-Hughes (Reviewer #3).

The reviewers have discussed the reviews with one another and the Reviewing Editor has drafted this decision to help you prepare a revised submission.

Summary:

Levendosky and Bowman describe analysis of the mechanism of ATP-dependent chromatin remodeling by the Chd1 and SNF2h remodelers, addressing the role of the acidic patch normally localized on the surface H2A/H2B dimers facing solution in the nucleosome. The Chd1 and ISWI remodelers participate in establishing the properly spaced chromatin structure. Using an elegant approach for formation of symmetrical and asymmetrical nucleosomes containing WT or mutant version of the acidic patch, the authors have shown convincingly that Chd1 and ISWI have distinct requirements for the acidic patch. Specifically, action of Chd1 nearly equally depends on the presence of the intact acidic patch at the entry and exit sides, while SNF2h much more strongly depends on the presence of entry-side acidic patch. Moreover, for SNF2h the requirement for the entry-side acidic patch is more important than the requirement for entry side DNA; this feature allows SNF2h to slide nucleosomes off DNA ends on asymmetric mutant nucleosomes containing mutant acidic patch at the exit side. Taken together, the data suggest existence of important functional differences between functionally similar Chd1 and ISWI remodelers.

Overall, this high-quality, technically outstanding work is addressing an important issue, the experimental findings are interesting and important, the data presented support the primary conclusions of the authors, and the manuscript is well written. However, we have some concerns that need to be addressed by the authors:

Essential revisions:

1) A limitation is that it is not known how SNF2h contacts the acidic patch, and it appears this interaction may involve a contact that normally serves to limit the extent of remodelling.

Cross-linking is used to great effect to reveal repositioning of histones that is not evident from native gels (APM/APM 80N0 nucleosomes). The WT/APM nucleosomes are partially repositioned by gel shift, so it is important that cross-linking is applied to determine the direction with which nucleosomes are repositioned in the presence of the of the WT/APM dimer combination.

2) Figure 2A: Some of the nucleosome preparations contain considerable amounts of hexasomes (especially 0N80 APM/ APM and 80N0 WT/APM samples). Hexasomes are likely remodeled at a different rate that is difficult to account for. Perhaps the experiments should be repeated for these samples using more homogeneous nucleosome preparations.

3) Figure 2B: It appears from the plots that nucleosome sliding occurs with 100% efficiency for all samples, but according to Figure 2A each sample contains a fraction of nucleosomes in the original position on the gel, variable between the samples. It seems that the authors normalized the plots, considering remodeling at saturation as 100%. This presentation is misleading because the differences in the efficiency of remodeling between the samples are not reflected in the plot. The data should be replotted to reflect the differences in the efficiency of remodeling between the samples.

4) Figure 3: To allow better comparison between the samples, the differences in the time scales should be emphasized. It is particularly important for low-rate reactions (e.g. WT/APM samples). The amounts of histone-free DNA should be quantified and quantitatively compared. Repositioning by SNF2h is about 4 times slower than observed with Chd1 in the presence of 5 times more enzyme. It is a concern that SNF2h is less active and not tested in the context of its normal interaction partners. If it is not possible to use ACF enzyme, the potential limitations of using SNF2h should be mentioned.

5) Figure 4: Markers used to calibrate the gels should be shown in the figure. Mapping of WT/WT nucleosome remodeling should also be shown.

6) The positions of both dimers (preferably and no-preferably binding to nucleosomal DNA) have to be shown on the sequence of the template, perhaps in the supplement. I did not find this description in this or in the original 2016 manuscript.

---

## [Author Response]

Essential revisions:1) A limitation is that it is not known how SNF2h contacts the acidic patch, and it appears this interaction may involve a contact that normally serves to limit the extent of remodelling.

We agree that determining the site of the acidic patch interaction is an important question in the field. Previous work by the Narlikar lab showed that the regulatory element NegC appeared to be necessary for responding to acidic patch mutations (Gamarra et al., 2018). Therefore, such an interaction is likely not critical for sliding activity per se, but instead relieves NegC or other inhibitory elements. Given the common inhibitory elements shared between Chd1 and ISWI remodelers (chromodomains and AutoN; the C-terminal bridge and NegC), we suspect that a key difference may be the strength or nature of interaction with the nucleosome acidic patch. While our manuscript does not shed light on how remodeler interactions may trigger release from autoinhibition, we felt it important to report how these two otherwise similar classes of remodelers respond so differently to nucleosomes with asymmetric availability of the wild type acidic patch.

Cross-linking is used to great effect to reveal repositioning of histones that is not evident from native gels (APM/APM 80N0 nucleosomes). The WT/APM nucleosomes are partially repositioned by gel shift, so it is important that cross-linking is applied to determine the direction with which nucleosomes are repositioned in the presence of the of the WT/APM dimer combination.

To address this point, we performed cross-linking with the other four combinations of WT and APM H2A/H2B dimers for both 0N80 and 80N0 nucleosomes. The four combinations for 0N80 nucleosomes (WT/WT, WT/APM, APM/WT, and APM/APM) are shown in Figure 4. The analogous combinations for 80N0 nucleosomes are shown in Figure 4—figure supplement 1. The question raised by the reviewers – what happens to the WT/APM combination – is answered by these experiments, which show that for asymmetric nucleosome with either orientation of mutant and wild type H2A/H2B dimers (APM/WT and WT/APM), the histone octamer is always shifted past the DNA end on the WT side. Thus, the location of the WT H2A/H2B dimer defines the preferential direction of nucleosome sliding, with the WT dimer on the entry side.

2) Figure 2A: Some of the nucleosome preparations contain considerable amounts of hexasomes (especially 0N80 APM/ APM and 80N0 WT/APM samples). Hexasomes are likely remodeled at a different rate that is difficult to account for. Perhaps the experiments should be repeated for these samples using more homogeneous nucleosome preparations.

In our previous paper describing the generation of asymmetric nucleosomes from oriented hexasomes (Levendosky et al., 2016), we studied how nucleosome sliding rates were effected by under- and over-saturating amounts of H2A/H2B dimer (Figure 7—figure supplement 1). By titrating different amounts of H2A/H2B dimer, we showed that nucleosome sliding rates are unaffected by hexasome, and in fact over-saturating amounts of dimer can reduce the observed rates and produce other species on native gels. We therefore routinely use an under-saturating amount of H2A/H2A to minimize the amount of misincorporated dimer, resulting in some hexasome, which should not impact the observed rates of nucleosome sliding.

3) Figure 2B: It appears from the plots that nucleosome sliding occurs with 100% efficiency for all samples, but according to Figure 2A each sample contains a fraction of nucleosomes in the original position on the gel, variable between the samples. It seems that the authors normalized the plots, considering remodeling at saturation as 100%. This presentation is misleading because the differences in the efficiency of remodeling between the samples are not reflected in the plot. The data should be replotted to reflect the differences in the efficiency of remodeling between the samples.

The reviewers are correct that the sliding rate plots were normalized. We have remade this graph so that now, for each nucleosome construct, the relative amount of shifted nucleosome is also shown.

4) Figure 3: To allow better comparison between the samples, the differences in the time scales should be emphasized. It is particularly important for low-rate reactions (e.g. WT/APM samples).

Yes, given the dramatic differences in sliding rates for different nucleosomes, different time scales were used for different reactions to better sample the progress curves. To draw attention to the timescale used for each reaction, we have highlighted the 1 min and 16 min time points above each gel, which are common to all reactions.

The amounts of histone-free DNA should be quantified and quantitatively compared.

The amount of free DNA is now plotted for Chd1, SNF2h and ACF (Figure 3C). This graph shows that free DNA was produced with APM/WT 0N80 nucleosomes when remodeled by SNF2h and ACF but not Chd1. A comparison to free DNA produced with WT/WT nucleosomes is also shown.

Repositioning by SNF2h is about 4 times slower than observed with Chd1 in the presence of 5 times more enzyme. It is a concern that SNF2h is less active and not tested in the context of its normal interaction partners. If it is not possible to use ACF enzyme, the potential limitations of using SNF2h should be mentioned.

As suggested by the reviewers, we have added nucleosome sliding experiments using ACF. With ACF, we observed the same behavior as SNF2h, even with a lower concentration of remodeler (100 nM instead of 1 µM). Thus, the difference between Chd1 and SNF2h is not simply due to the higher concentrations required or an absence of the Acf1 auxiliary subunit. This material was purchased and extremely limiting, precluding histone mapping experiments but sufficient for native gel sliding experiments.

5) Figure 4: Markers used to calibrate the gels should be shown in the figure. Mapping of WT/WT nucleosome remodeling should also be shown.

The sequencing ladders used as markers are shown in Figure 4—figure supplements 2 and 3. Mapping of WT/WT nucleosomes is now included as requested in Figure 4 and Figure 4—figure supplement 1.

6) The positions of both dimers (preferably and no-preferably binding to nucleosomal DNA) have to be shown on the sequence of the template, perhaps in the supplement. I did not find this description in this or in the original 2016 manuscript.

The location of the histone-fold contacts to the Widom 601 are now depicted in Figure 1—figure supplement 1.